# Spatial distribution and determinants of improved shared sanitation facilities among households in Ethiopia: Using 2019 mini-Ethiopian Demographic and Health Survey

**Baye Tsegaye Amlak**[1]*, **Daniel Gashaneh Belay**[2,3]

1 Department of Nursing, College of Medicine and Health Sciences, Debre Markos University, Debre Markos, Ethiopia, 2 Department of Human Anatomy, School of Medicine, College of Medicine and Health Sciences, University of Gondar, Gondar, Ethiopia, 3 Department of Epidemiology and Biostatistics, Institute of Public Health, College of Medicine and Health Sciences, University of Gondar, Gondar, Ethiopia

* baye_tsegaye@dmu.edu.et

**Data Availability Statement:** The data utilized in this study were accessed from the Demographic and Health Surveys (DHS) Program database

## Abstract

### Introduction

Limited or shared sanitation services are considered improved sanitation facilities, but they are shared between two or more households. Globally, 600 million people use shared toilet facilities. Although shared facilities are not classified as improved sanitation due to potential infection risks, inaccessibility, and safety concerns, this is a significant issue in developing countries like Ethiopia. Evidence on the distribution of shared sanitation services and their determinants in Ethiopia is limited. Therefore, this study aimed to assess the extent of shared toilet facilities and their determinants among households in Ethiopia.

### Methods

The 2019 Ethiopian Demographic and Health Survey (EDHS) served as the basis for the cross-sectional secondary data analysis. The analysis included a total of 7,770 households from the weighted sample. STATA 14 software was used to clean, weigh, and analyze the data. To explore the distribution and determine the factors associated with shared toilet facilities in Ethiopia, both spatial and mixed-effect analyses were utilized. A p-value of less than 0.05 was used to display the relationships between the dependent and independent variables, employing adjusted odds ratios and 95% confidence intervals.

### Results

The magnitude of improved shared sanitation facilities among households in Ethiopia, according to the EDHS 2019, was 10.5% (95% CI: 9.88, 11.24). The prevalence was highest in Addis Ababa at 70.2% and lowest in the Southern Nations, Nationalities, and Peoples' Region at 2.4%. Individual-level variables significantly associated with the use of improved shared toilet facilities included being a household head aged 55 years or older [AOR = 0.48;

following a formal registration process. Interested researchers can obtain the data by registering and submitting a request via the DHS Program's website at https://dhsprogram.com/data/new-user-registration.cfm.

**Funding:** The author(s) received no specific funding for this work.

**Competing interests:** The authors declare that they have no competing interests.

**Abbreviations:** AOR, Adjusted Odds Ratio; CI, Confidence Interval; CSA, Central Statistical Agency; EDHS, Ethiopian Demographic and Health Survey; HR, Household Record; MOH, Ministry Of Health; OD, Open Defecation; SDG, Sustainable Development Goal.

95% CI: 0.33, 0.71], having secondary education or higher [AOR = 2.43; 95% CI: 1.80, 3.28], and belonging to middle or rich wealth status [middle: AOR = 2.32; 95% CI: 1.35, 3.96; rich: AOR = 6.23; 95% CI: 3.84, 10.11]. Community-level characteristics such as residing in urban areas [AOR = 7.60; 95% CI: 3.47, 16.67], the metropolitan region [AOR = 25.83; 95% CI: 10.1, 66.3], and periphery regions [AOR = 5.01; 95% CI: 2.40, 10.48] were also associated with the use of shared toilet facilities.

## Conclusion

The usage of improved shared toilet facilities among households in Ethiopia is relatively low. Significant factors related to the use of shared toilet facilities were being 55 years of age or older, possessing secondary or higher education, having a middle or rich wealth status, living in urban areas, and residing in metropolitan or peripheral regions. To improve access to and utilization of shared sanitation facilities, Ethiopian policy should emphasize user education and awareness.

## Introduction

The World Health Organization (WHO) and the United Nations Joint Monitoring Programme (JMP) define limited sanitation services as improved sanitation facilities that are shared between two or more households [1, 2]. Globally, 2.3 billion people lack access to improved sanitation facilities [2]. From these, 600 million people use shared toilet facilities [2].

Safely managed sanitation is the only option available to millions of people living in densely packed urban areas, especially in informal settlements. This is preferable to open defecation, which has far more severe negative impacts on health, safety, and dignity [1]. However, while shared toilets are designed to be an improved type of sanitation facility, they are considered unimproved because they are shared among multiple users [3].

A meta-analysis of global studies on shared sanitation in informal settlements estimated that the overall prevalence of shared sanitation was 67%. Users' preferences for using shared facilities were influenced by factors such as cleanliness, affordability, safety, privacy, structural quality, and accessibility [4]. However, informal settlements are widespread and often characterized by substandard housing, poverty, and a lack of basic sanitation facilities. This is especially true in developing nations like Ethiopia, where conflict and internal displacement are prevalent [5]. Therefore, sharing toilets provides residents who lack private toilets in their homes with access to sanitation facilities [5].

In Sub-Saharan African countries, the number of households using shared toilet facilities increased from 0.64 million to 0.96 million, representing 0.08% of all households [2, 6]. Similarly, in Ethiopia, the use of shared toilet facilities rose from 4% to 7% between 2000 and 2015 [7]. Shared facilities are not regarded as an improvement in sanitation because they frequently encounter maintenance issues. Furthermore, their accessibility can lead to the spread of infections due to poor hygiene, limited accessibility, and unsafe conditions [8]. These are frequently not well kept, creating unclean conditions that discourage frequent use [1]. Global research has shown a connection between the use of shared toilets and adverse health outcomes such as helminth infections, diarrhea, enteric fevers, and fecal-oral diseases [9]. It aids the transmission of microorganisms that cause diarrheal diseases [10], with children being the most vulnerable [11]. Diarrheal disease is the second leading cause of death for children under five

worldwide, accounting for 760,000 deaths and 1.7 million morbidities annually [12]. In Africa, diarrhea is one of the main causes of death in under-five children [13], It causes the deaths of half a million children under the age of five annually in Ethiopia alone [14]. Moreover, using shared toilets puts hundreds of millions of women and children at greater risk of sexual exploitation and a lack of privacy during their menstrual cycles [5, 15]. Poor sanitation is also associated with infections and eye diseases, such as trachoma [5].

Sustainable Development Goal (SDG) target 6.2 aims to ensure that by 2030, all children will have access to universal sanitation facilities [8] and that no child should suffer from disease or die due to contaminated water or contact with human waste [16]. Therefore, providing shared sanitation services could be a crucial initial step toward achieving the universal sanitation coverage goal set out in the Sustainable Development Goals [8]. Since 1995, Ethiopia has prioritized its sanitation program, following the inclusion of public health in the country's National Constitution. Subsequently, in 2005 and 2006, the Ministry of Health developed the National Hygiene and Sanitation Strategy and the National Hygiene and On-Site Sanitation Protocol, respectively [17, 18]. However, the magnitude and contributing factors to using shared toilet facilities in Ethiopia are little known [19]. Therefore, this study aims to address the following research questions: What is the magnitude and spatial distribution of improved shared sanitation facilities in Ethiopia? What factors are associated with improved shared sanitation facilities? Answering these questions will provide valuable insights for policymakers and program planners, helping them allocate resources more effectively, design targeted interventions, and develop relevant policies.

## Methodology

### Study design, setting, and data source

This study utilized data from the recent Ethiopian Demographic and Health Survey (EDHS 2019), which was collected using a community-based cross-sectional study design. Since 1984, the DHS has gathered a broad range of objective and self-reported data across more than 99 countries. Key advantages of the DHS include high response rates, national coverage, rigorous interviewer training, standardized data collection procedures across countries, and consistent content over time [20].

With 1.1 million square kilometers under its belt and an expected 132,059,767 inhabitants in 2024, Ethiopia is the second most populated country in Africa next to Nigeria [21]. Ethiopia has two city administrations (Addis Ababa and Dire Dawa) and nine regions (Tigray, Afar, Amhara, Oromia, Benishangul Gumuz, Somalia, South Nation Nationalities and Peoples of Ethiopia (SNNP), Gambelia, and Harari) with a federal decentralized administrative structure. To ensure survey precision was comparable across regions, the sample allocation was carried out through equal distribution, with 35 enumeration areas (EAs) selected from each of the three larger regions: Amhara, Oromia, and the Southern Nations, Nationalities, and Peoples' Region (SNNPR). Additionally, 25 EAs were selected from each of eight other regions. In the first stage, a total of 305 EAs (93 in urban areas and 212 in rural areas) were chosen with probability proportional to EA size. In the second stage, an average of 25–30 households were carefully selected from each EA based on the 2019 Ethiopian Population and Housing Census (EPHC) frame [22].

### Study population

The study population consisted of all houses that had unimproved sanitation services evaluated during the 2019 mini EDHS survey (7,561). This includes, shared improved sanitation services (1,276), unimproved sanitation facilities (3,442), and open defecation (OD) (2,843).

Out of the 8,663 households included in the 2019 mini EDHS survey, 1,102 households had improved but not shared sanitation facilities and were excluded from the analysis. Ultimately, a sample of 7,561 households (weighted to 7,770) was included in the analysis.

## Study variables

The outcome variable of the study was shared/ limited sanitation service which means households with either one type of improved sanitation facility but shared with other households [23] (**Table 1**).

The study's predictor variables were categorized into individual-level variables, which included the age, sex, and educational attainment of household heads, as well as factors such as the household wealth index, family size, and household size. Additionally, community-level characteristics such as residence location, region, and community poverty were examined. After assessing the normal distribution of the aggregated community components using a histogram and the Shapiro-Wilk test, the data were recorded using the appropriate measures of central tendency (**Table 2**).

## Data management and analysis

This study utilized data from the EDHS 2019 provided by the DHS program. The results and independent variables were extracted using the household data (HR) set, and STATA version 14 was employed for recording, extracting, and analyzing the data. To ensure the representativeness of the survey and obtain reliable estimates and standard errors, the data were weighted for sampling probabilities using the weighting factor before performing any statistical analysis.

Multilevel analysis was used to account for both individual and community levels due to the hierarchical structure of the EDHS data, where households are nested within enumeration areas (EAs). This approach addresses the violation of the assumption of independence of observations and equal variance across clusters. The Interclass Correlation Coefficient (ICC), Median Odds Ratio (MOR), and Proportional Change in Variance (PCV) were utilized to measure variance through random effects.

The ICC reveals the variation of shared toilet facilities usage between clusters is calculated as; $ICC = \frac{VC}{VC+3.29}*100\%$, where; VC = cluster level variance.

The MOR is defined as the median value of the odds ratio between the area at the lowest risk and the highest risk of shared toilet facilities usage when randomly picking out two clusters.

$MOR = e^{0.95\sqrt{VC}}$ Where; VC is the cluster-level variance.

**Table 1. Questions to measure improved but shared/ limited sanitation service.**

| Outcome variable (Shared/ limited sanitation service) | Control group/comparators |
| --- | --- |
| Did the toilet have flush/pour flush to the piped sewer system? | ✓ Unimproved sanitation services such as |
| Did the toilet have flush/pour flush to the septic tank? | ◦ Pit latrine without slab/ open pit |
| Did the toilet have a flush to somewhere else? | ◦ Bucket toilet |
| Did the toilet have flush/pour flush to the pit latrine? | ◦ Hanging toilet/latrine |
| Did the toilet have a flush, *don't know where*. | ◦ Other |
| Was the toilet Ventilated Improved Pit (VIP) latrine? | ✓ Open defecation: no facility/ bush/field |
| Was the toilet pit latrine with a slab? | |
| Was the toilet composting toilet? | |
| If a household uses any of the listed above-improved sanitation facilities but shares them with other households. | |

**Table 2. A list of the study's variables along with an explanation of each measurement.**

| Level | Variables | Measurements |
|---|---|---|
| Individual level variables | Age | The age of the participants was categorized as 15–24, 25–40, 41–54, >55 |
| | Sex | Sex of the household head as male and female |
| | Education level | Educational attainment is categorized as uneducated, primary, secondary, and above |
| | Family size | Categorized as 1–3, 4–6, and 7 and above. |
| | Wealth index | A wealth index categorized as poorest, intermediate, richest, and wealthiest in the DHS data collection was developed using principal components analysis and included in the datasets. We classified it into three categories for this study: middle class, rich (including wealthier), and poor (including poorer and poorest). |
| Community level variables | Residency | Urban or rural based on where the household lives in the dataset was used without change. |
| | Region | Ethiopia's eleven regions are divided into three groups according to their level of development and need for government assistance: the "three metropolises" (Addis Ababa, Harari, and Diredewa); the "large central" (Tigray, Amhara, Oromia, SNNPR); and the "small peripherals" (Afar, Benshangul-Gumuz, Gambelia, and Somali) [24]. |
| | Community level poverty | The percentage of households in the lowest and poorest quintiles found in the wealth index data was used to calculate the community's level of poverty. classified as high if the percentage of households falling into the poor categories exceeded 50% and as low if the percentage was less than 50%. |

The PCV shows the variation in shared toilet facilities usage among households explained by both individual and community level factors. $PCV = \frac{Vnull - VC}{Vnull} * 100\%$ Where; Vnull = variance of the initial model, and VC = cluster level variance of the next model [25–27].

Generally, four models were fitted in multi-level analysis. The first model was the null model, which simply included the outcome variable and was meant to examine the cluster's variability in shared toilet facility utilization. Household and community-level factors are included in the second and third multilevel models, respectively, while shared toilet facility usage was simultaneously fitted to both household and community-level variables in the fourth model. The deviation test was used to compare the models, and the model that suited the data the best was the one with the lowest deviance [25–27]. In the multivariable analysis, the associations between dependent and independent variables were presented using adjusted odds ratios and 95% confidence intervals with a p-value of <0.05.

## Spatial analysis

The Global Moran's I statistic was used to assess spatial autocorrelation [28]. The Global Moran's I value ranges from −1 to +1, where a value below 0 indicates negative spatial autocorrelation, and values above 0 indicate positive spatial autocorrelation [28, 29]. Based on sampled clusters, we employed a spherical semivariogram ordinary Kriging-type spatial interpolation technique to forecast the extent of shared toilet facilities in Ethiopia for unsampled areas. The input for spatial prediction was the percentage of shared restrooms in each cluster. To determine the locations of shared restroom clusters, Bernoulli-based model spatial scan statistics were applied [30]. To suit the Bernoulli model, the houses without shared toilets were taken as controls, and the scanning window that moves over the research region with shared toilet facilities was taken as a case.

### Ethical approval statement

The study doesn't involve the collection of information from subjects, secondary data analysis was done. Ethical approval and consent to participate are not applicable. Since the study is a secondary data analysis based on DHS data.

## Results

### Sociodemographic characteristics of the study participants

This study included a total of 7,770 households. Among them, males constituted more than three-fourths (6,046 or 77.81%) of the household heads. The majority of participants (72.19%) lived in rural areas, and nearly half (3,785 or 48.71%) of the household heads had no formal education.

As the age of household heads increased from 13–30 years to above 57 years, the use of shared toilet facilities decreased from 20.35% to 6.9%, respectively. More than two-thirds (69.09%) of households in metropolitan regions and about one-fourth (28.95%) of household heads in urban residences used shared toilet facilities [Table 3].

### The magnitude of shared sanitation facilities in Ethiopia

In the EDHS 2019, the magnitude of improved shared sanitation facilities among households in Ethiopia was 10.5% (95% CI: 9.88, 11.24). Of these, more than two-thirds (69.5%) used

**Table 3. Socio-demographic characteristics and improved shared toilet usage in Ethiopia, mini 2019 EDHS.**

| Variables | Categories | Shared toilet facilities | | Total weighted frequency (%) |
|---|---|---|---|---|
| | | Yes (%) | No (%) | |
| | | n = 819 (10.5) | n = 6,951 (89.5) | |
| Age of household head (years) | 15–24 | 184 (20.35) | 722 (79.65) | 906 (11.66) |
| | 25–40 | 370 (11.40) | 2875 (88.60) | 3245 (41.75) |
| | 41–54 | 136 (7.74) | 1616 (92.26) | 1751 (22.53) |
| | >55 | 129 (6.92) | 1739 (93.08) | 1868 (24.05) |
| Sex of household head | Male | 501 (8.28) | 5545 (91.72) | 6,046 (77.81) |
| | Female | 318 (18.470 | 1406 (81.53) | 1,724 (22.19) |
| Educational attainment of household head | No education | 228 (6.01) | 3,556 (93.99) | 3,785 (48.71) |
| | Primary education | 244(8.79) | 2,530 (92.21) | 2,775(35.71) |
| | Secondary & above | 348 (28.71) | 863 (71.29) | 1,211 (15.58) |
| Household family size | 1–3 | 49 8 (17.96) | 2141 (82.04) | 2,610 (33.59) |
| | 4–6 | 249 (7.35) | 3140 (92.65) | 3,889 (43.61) |
| | 7 & above | 101 (5.72) | 1670 (94.28) | 1,771 (22.80) |
| Wealth index | Poor | 60 (2.01) | 2,929 (97.99) | 2,989 (38.47) |
| | Middle | 36 (2.35) | 1,516 (97.65) | 1,553 (19.98) |
| | Rich | 723 (22.39) | 2,506 (77.61) | 3,229 (41.55) |
| **Community level variables** | | | | |
| Residence | Urban | 625 (28.95) | 1,535 (71.05) | 2,161 (27.81) |
| | Rural | 193 (3.45) | 5,415 (96.55) | 5,609 (72.19) |
| Region | Metropolis | 173 (69.09) | 77 (30.91) | 250 (3.22) |
| | Large centrals | 530 (7.68) | 6,373 (92.32) | 6,904 (88.85) |
| | Small periphery | 116 (18.82) | 500 (81.18) | 615 (7.93) |
| Community poverty level | Low | 711 (15.43) | 3,898 (84.57) | 4,610 (59.33) |
| | High | 108 (3.41) | 3,052 (96.59) | 3,160 (40.67) |

shared pit latrines with slabs, while only 0.6% used shared sanitation facilities connected to a flush-piped sewer system. The highest prevalence of shared toilet facilities was observed in Addis Ababa (70.2%), whereas the lowest was in the Southern Nations, Nationalities, and Peoples' Region (2.4%) (**Fig 1**).

## Factors associated with shared or limited access to improved sanitation service in Ethiopia

Since the models were nested, log-likelihood and deviance tests were performed for model comparison. The III-level binary logistic regression model was chosen because it had the largest LR (-1438) and the smallest deviance test result (2,876). The random effects of ICC, PCV, and MOR were evaluated.

The difference at the cluster level was responsible for almost 79% of the variability in improved shared sanitation facilities among sample families, according to the ICC in the null model. According to the MOR value in the null model (28.75), a household from a high-risk cluster had 28.75 times higher odds of having shared toilet facilities compared to a household from a low-risk *cluster*. Additionally, the PCV value in the final model demonstrated that characteristics at both the individual and community levels simultaneously explained nearly 76% of the variation in improved shared toilet facilities among study families [**Table 4**].

Based on the results from Model III, there was a positive association between improved shared sanitation facilities and several factors such as individuals with higher educational attainment and from greater household wealth, those living in urban areas and metropolitan regions.

As the age of household heads increased to 15–24 years, 25–40 years, and over 55 years, the odds of using improved shared toilet facilities decreased by 33%, 46%, and 52%, respectively [AOR = 0.66; 95% CI: 0.50, 0.89], [AOR = 0.54; 95% CI: 0.37, 0.79], and [AOR = 0.48; 95% CI: 0.33, 0.71]. The odds of using improved shared toilet facilities were doubled among household heads with above-primary education [AOR = 2.43; 95% CI: 1.80, 3.28].

Households with a middle or rich wealth index were 2 and 6 times more likely to use improved shared toilet facilities compared to poor households [AOR = 2.32; 95% CI: 1.35, 3.96] and [AOR = 6.23; 95% CI: 3.84, 10.11], respectively. Individuals living in urban areas were 7.6 times more likely to use improved shared toilet facilities compared to those in rural

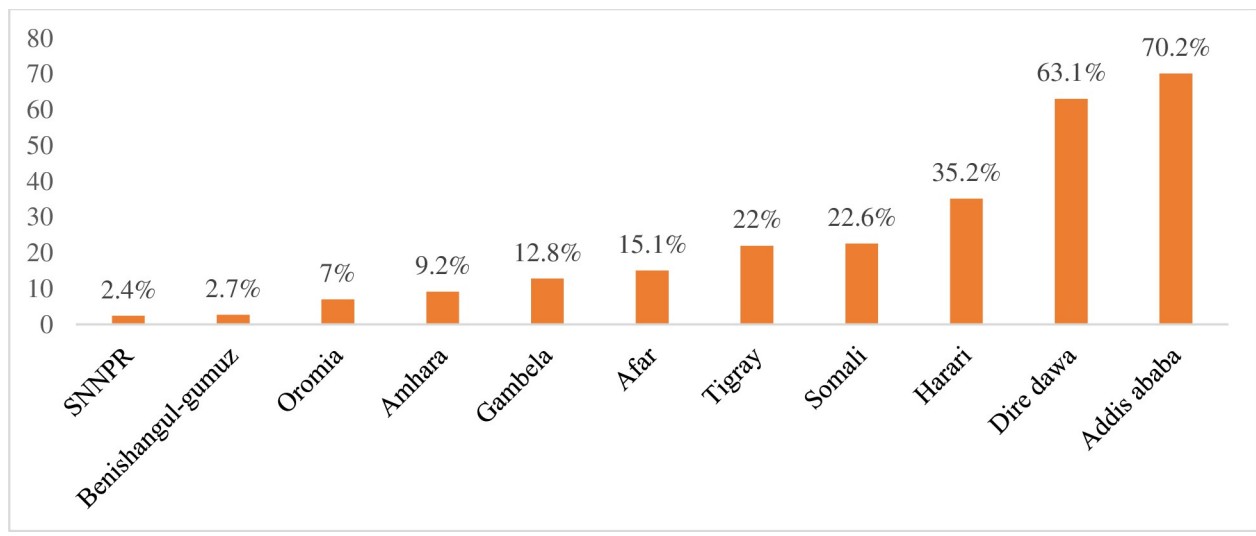

**Fig 1. Magnitude of improved shared toilet facilities among households in Ethiopia, using the 2019 EDHS.**

**Table 4. Multilevel analysis of factors associated with improved shared toilet facilities usage among households in Ethiopia.**

| Variables | Categories | Null model | Model I | Model II | Model III |
|---|---|---|---|---|---|
| | | | AOR [95% CI] | AOR [95% CI] | AOR [95% CI] |
| Age of household head (years) | 15–24 | | 1.00 | ------------ | 1.00 |
| | 25–40 | | 0.67 [0.50, 0.91]* | ------------ | **0.66 [0.50, 0.89]*** |
| | 41–54 | | 0.56 [0.38, 0.81]** | ------------ | **0.54 [0.37, 0.79]*** |
| | >55 | | **0.49 [0.34, 0.72]*** | ------------ | **0.48 [0.33, 0.71]*** |
| Sex of household head | Male | | 1.00 | ------------ | 1.00 |
| | Female | | 1.17 [0.86, 1.37] | ------------ | 1.23 [0.92, 1.47] |
| Educational attainment of household head | No education | | 1.00 | ------------ | 1.00 |
| | Primary education | | 1.36 [0.99,1.78] | ------------ | 1.31 [0.98, 1.73] |
| | Secondary& above | | **2.54 [1.88, 3.44]** | ------------ | **2.43 [1.80, 3.28]*** |
| Household family size | 1–3 | | 1.00 | ------------ | 1.00 |
| | 4–6 | | 0.75 [0.61, 1.03] | ------------ | 0.76 [0.60, 1.01] |
| | 7 & above | | 0.88 [0.62, 1.26] | ------------ | 0.97 [68, 1.38] |
| Wealth index | Poor | | 1.00 | ------------ | 1.00 |
| | Middle | | 2.66 [1.55, 4.56]** | ------------ | **2.32 [1.35, 3.96]*** |
| | Rich | | **10. 25 [6.39, 16.41]** | ------------ | **6.23 [3.84, 10.11]*** |
| **Community level variables** | | | | | |
| Residence | Urban | | ------------ | **14.6 [6.61, 32.21] *** | **7.60 [3.47, 16.67]*** |
| | Rural | | ------------ | 1.00 | 1.00 |
| Region | Metropolis | | ------------ | 28.97 [11.21, 74.87]*** | **25.83 [10.1, 66.3] *** |
| | Small periphery | | ------------ | **4.51 [2.14, 9.49]*** | **5.01 [2.40, 10.48] *** |
| | Large central | | ------------ | 1.00 | 1.00 |
| Community poverty usage | Low | | ------------ | 1.00 | 1.00 |
| | High | | ------------ | 0.23 [0.11, 0.49] | 0.65 [0.29, 1.39] |
| **Random effect** | | | | | |
| | Variance | 12.5 [8.99, 17.46] | 6.75 [4.84, 9.41] | 3.27 (2.34, 4.55) | 3.05 [2.17, 4.29] |
| | ICC | 0.79 [0.73, 0.84] | 0.67 [0.59, 0.74] | 0.50 [0.42, 0.58] | 0.48 [0.40, 0.57] |
| | MOR | 28.75 | 11.80 | 5.57 | 5.25 |
| | PCV | Reff | 0.46 | 0.74 | 0.76 |
| **Model comparison** | | | | | |
| | Log-likelihood | -1654 | -1515 | -1544 | -1438 |
| | Deviance | 3.308 | 3,030 | 3,088 | 2,876 |
| | Mean VIF | --- | 1.59 | 2.18 | 1.96 |

* = P-value < 0.05

** = Pvalue < 0.01

*** = Pvalue < 0.001

ICC = Inter cluster correlation coefficient, MOR = Median odds ratio, PCV = proportional change in variance. AOR = adjusted odds ratio; CI = confidence intervale,
VIF = Variance Inflation Factors

areas [AOR = 7.60; 95% CI: 3.47, 16.67]. Residents of metropolitan regions were 26 times more likely, and those in small periphery regions were 5 times more likely to use improved shared toilet facilities compared to those in large central regions [AOR = 25.83; 95% CI: 10.1, 66.3] and [AOR = 5.01; 95% CI: 2.40, 10.48], respectively [**Table 4**].

## Spatial analysis of improved shared toilet facilities utilization among households in Ethiopia

**Spatial autocorrelation analysis of improved shared toilet facilities.** Ethiopia's improved shared sanitation services spatial autocorrelation data revealed a strong positive spatial autocorrelation throughout the nation's regions. It was discovered to be grouped with the value of the Global Moran's Index: 1.78 with (p< 0.001) (**Fig 2**).

**Hotspot analysis of improved shared toilet facilities among household in Ethiopia.** The Getis-Ord Gi* hotspot analysis showed that improved shared toilet facilities were more practised in Addis Ababa, and Dire Dawa, whereas the SNNPR (South Nations Nationalities and People's Region), and Beneshangul Gumuz regions were the cold spot areas (**Fig 3**).

**Significant windows and SaTscan analysis of improved shared toilet facilities among households in Ethiopia.** The SaTScan analysis of improved shared toilet facilities among households in Ethiopia identified 86 primary clusters and 29 secondary clusters. The primary clusters were centered at coordinates 8.771915 N, 40.335915 E, with a radius of 206.97 km, and were located in Addis Ababa, northern parts of Oromia, southern parts of Amhara and Afar, and Dire Dawa. Households within these primary clusters were 5 times more likely to use improved shared toilet facilities compared to those outside these clusters (RR = 5.0, P-value < 0.001) (**Table 5** and **Fig 4**).

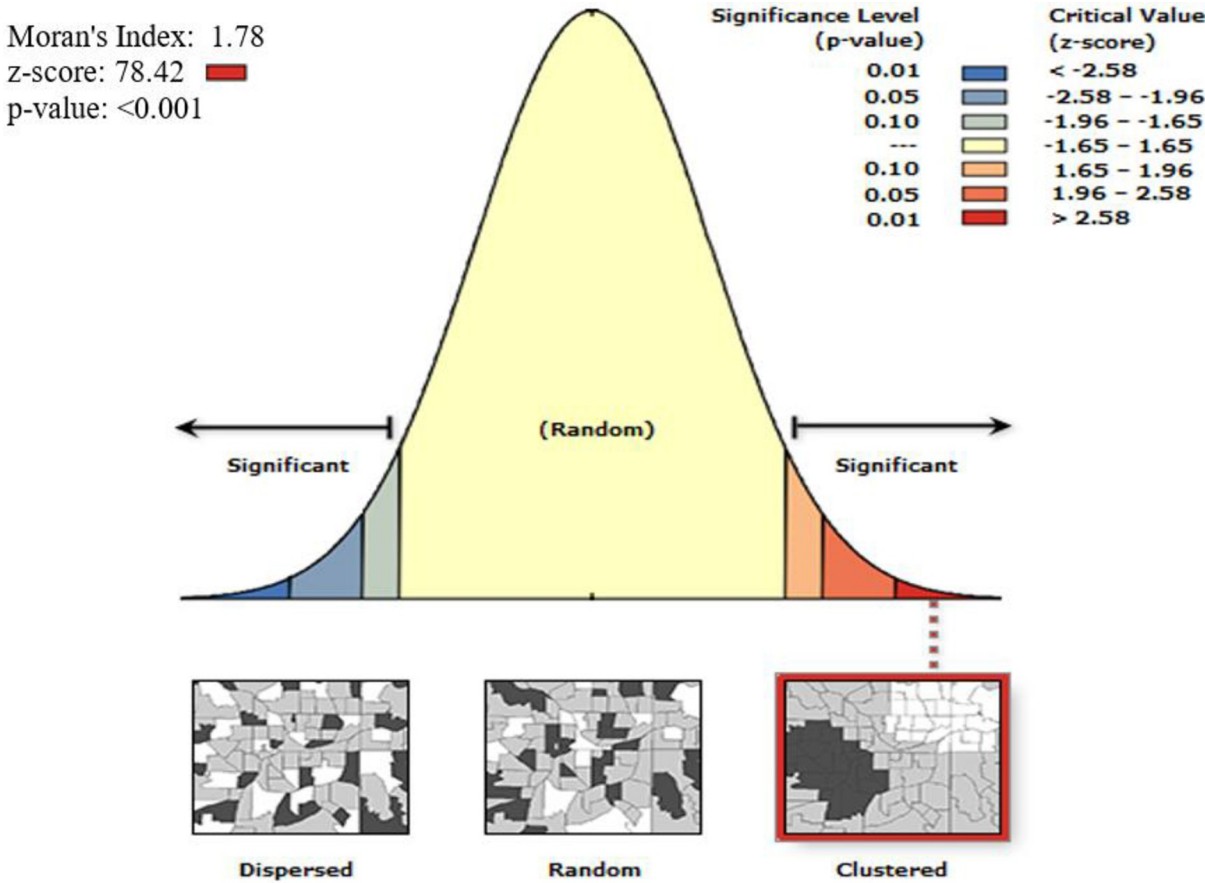

**Fig 2. Spatial autocorrelation analysis of improved shared toilet facilities among households in Ethiopia using 2019 mini-EDHS.** The base map for the shapefile was sourced from: https://gadm.org/.

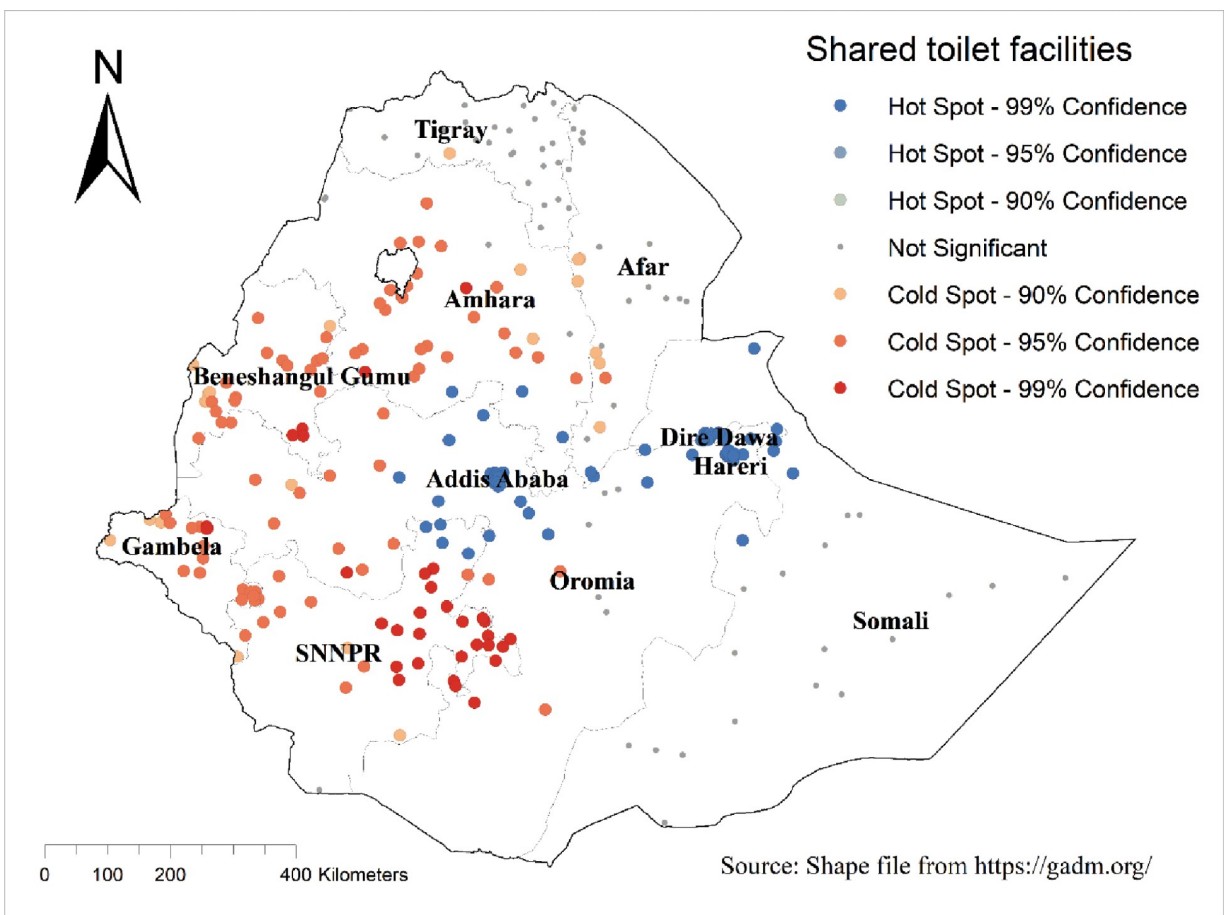

**Fig 3. Hotspot analysis of improved shared toilet facilities among households in Ethiopia using 2019 mini-EDHS.** The base map for the shapefile was sourced from: https://gadm.org/.

**Kiringing interpolation of improved shared toilet facilities among households in Ethiopia.** The Kriging interpolation method for improved shared toilet facilities among households in Ethiopia indicated that high-risk areas, such as Addis Ababa and Dire Dawa, had predicted usage rates ranging from 50% to 61%. Conversely, the lowest predicted usage rates were observed in the SNNPR, Gambela, Beneshangul Gumuz, and Oromia regions, ranging from 0% to 12% (**Fig 5**).

## Discussion

This study was conducted to assess the magnitude of improved shared toilet facilities and their determinants among households in Ethiopia. Based on this, the prevalence of improved shared toilet facilities in Ethiopia was 10.5% (95% CI: 9.88, 11.24). This is higher than a study by the WHO and UNICEF Joint Monitoring Program in Ethiopia (7%) [7]. It is also higher than in Yemen (4%) and Eritrea (5%), but lower than the global estimate reported in meta-analyses (67%) [4], in Kenya (21%), Ghana (57%), and Uganda (14%) [7]. This variation may be attributed to the different community initiative programs that employ more effective approaches to reducing unimproved sanitation practices and achieving desired sanitation outcomes [18, 31].

In this study, as the age of household heads increases, the usage of improved shared toilet facilities among households decreases. This is consistent with a study conducted in India,

**Table 5. Significant spatial clusters of improved shared toilet facilities among households in Ethiopia using 2019 mini-EDHS.**

| Clusters | Enumeration areas (clusters) detected | Coordinate/radius | Population | Cases | RR | LLR | P-value |
|---|---|---|---|---|---|---|---|
| 1ry (86) | 105, 88, 28, 41, 102, 106, 42, 127, 40, 104, 69, 90, 101, 43, 103, 108, 272, 269, 268, 271, 110, 280, 278, 273, 279, 270, 267, 264, 266, 277, 305, 275, 276, 265, 263, 256, 274, 304, 258, 257, 261, 260, 262, 50, 259, 303, 281, 296, 287, 68, 282, 302, 286, 284, 288, 283, 111, 294, 285, 291, 292, 293, 295, 297, 290, 289, 175, 231, 233, 246, 244, 232, 243, 237, 234, 236, 301, 298, 235, 242, 247, 241, 240, 253, 238, 239 | 8.771915 N, 40.335915 E / 206.97 km | 2447 | 851 | 5.04 | 181.4– 489.78 | <0.001 |
| 2nd (29) | 281, 282, 284, 283, 287, 285, 286, 288, 296, 291, 297, 292, 294, 290, 289, 295, 302, 264, 273, 267, 263, 270, 276, 265, 275, 266, 258, 271, | 10.589922 N, 34.352539 E / 88.60 km | 505 | 245 | 2.28 | 7.0– 181.3 | <0.001 |

which found that unimproved sanitation practices decrease among older household members [32]. This trend may reflect that as individuals age, they may experience disabilities or incontinence, making it more challenging to use outdoor or shared sanitation facilities [32]. Another factor could be that older individuals, on average, may have reduced mobility and greater difficulty moving freely outside their homes to access sanitation facilities.

In this study, as the educational status of household heads increased, the odds of using shared toilet facilities also increased. This is because one-third (32.23%) of households in Ethiopia use open defecation which is more prevalent in less socioeconomic regions [33]. Individuals with at least a formal education are more likely to use improved shared toilet facilities,

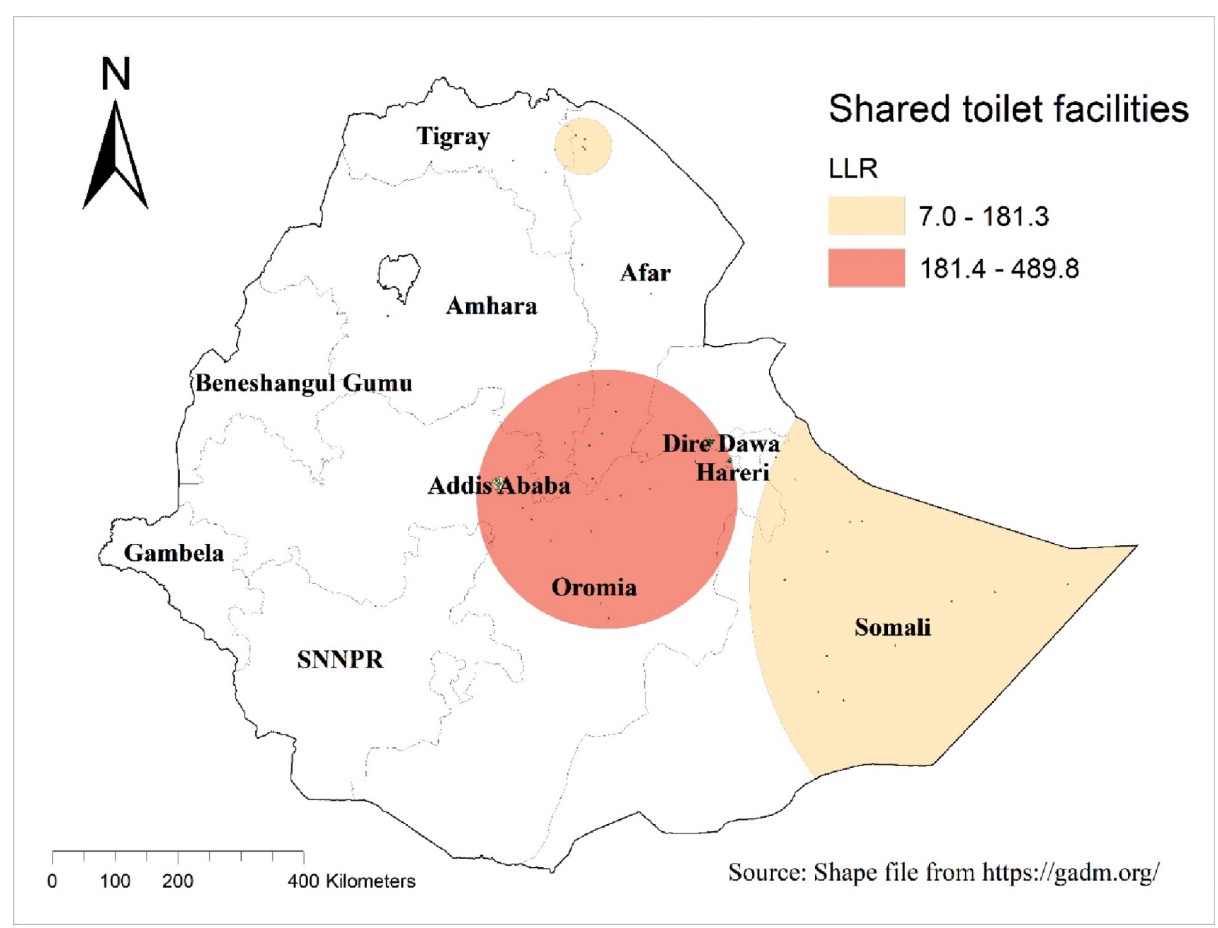

**Fig 4. SaTscan analysis of improved shared toilet facilities among households in Ethiopia using 2019 mini-EDHS.** The base map for the shapefile was sourced from: https://gadm.org/.

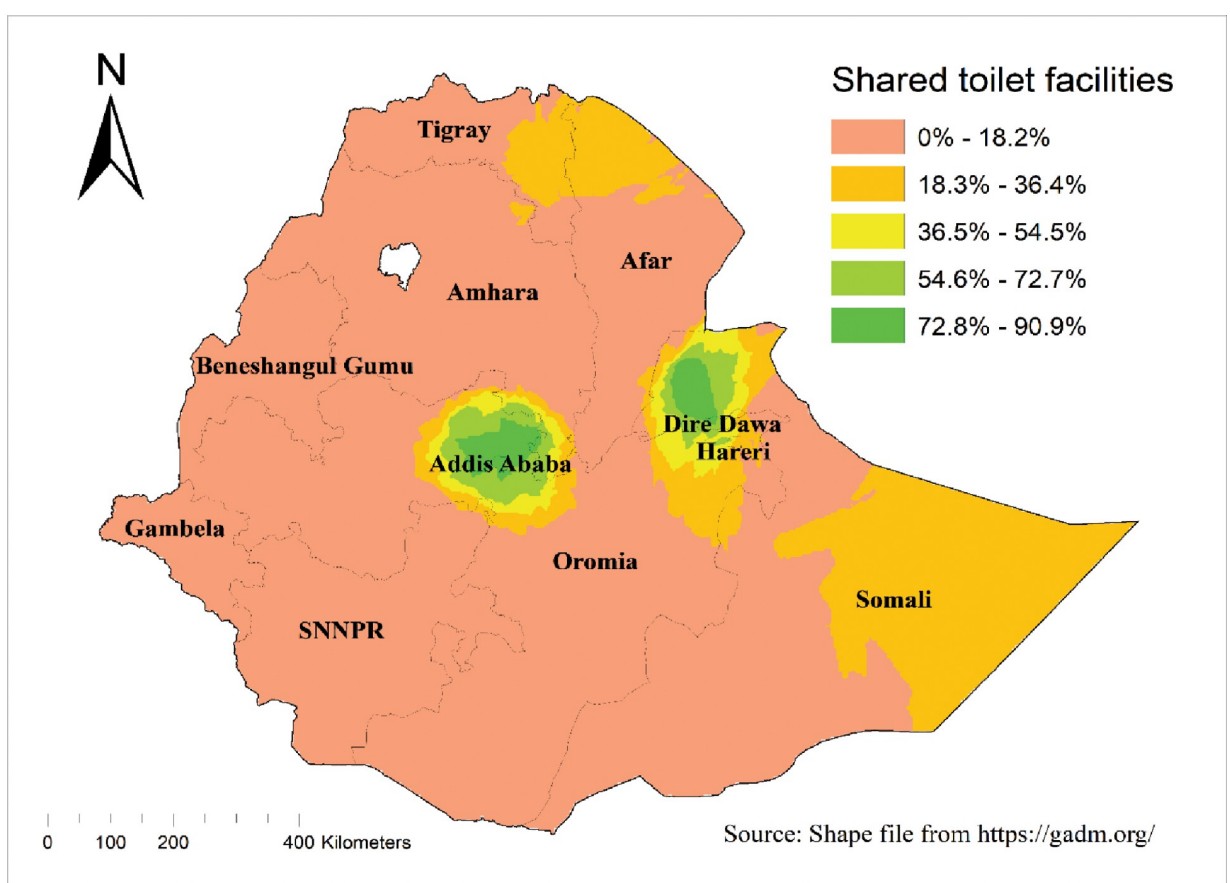

**Fig 5. Kiringing interpolation of improved shared toilet facilities among households in Ethiopia using 2019 mini-EDHS.** The base map for the shapefile was sourced from: https://gadm.org/.

which generally have fewer severe consequences for health, safety, and dignity compared to open defecation [1]. Furthermore, educated household heads are more likely to recognize the importance of having sanitation facilities, even if they are limited or shared [34, 35]. According to the World Health Organization report, education can raise the community's demand for better sanitation facilities [36]. Moreover, a higher level of education enhances awareness and fosters positive attitudes toward choosing relatively more upgraded sanitation facilities [37]. Specifically, educated women are more likely to prefer safe and sanitary facilities that offer privacy and maintain good quality during their menstrual cycle [38]. Moreover, well-educated households with higher incomes had better access to upgraded sanitation facilities [39].

Households that have a rich wealth index are more likely to use the improved shared toilet as compared to poor households. This is in line with Ethiopia [4], Nigeria [40], and Ghana [41]. This might be because the majority of improved shared toilet facilities are found in urban areas with a good income-earning population group as compared to open defecation practices that have been taking place in rural areas of low-income countries [42]. In addition, poor households may not have enough place and capacity to construct private and improved shared toilets and their only option is open defecation. Furthermore, poor households found in developing countries are mainly housed in slums that lack essential infrastructure [38].

The study also found that individuals living in rural households were less likely to use improved shared toilet facilities compared to those in urban areas. This finding is consistent

with other research indicating that shared toilet facilities are less common in rural areas, where open defecation remains prevalent, affecting 37% of the rural population in Ethiopia [43]. This may be because many governments do not prioritize rural sanitation in their national agendas, often lacking progressive budgetary support as well as essential legislative and institutional reforms [44]. Besides, the previous study findings suggested that rural households' willingness to pay for 'improved' latrines is minimal [45]. Moreover, rural households are uneducated, and they believe that open defecation is a routine sanitation service [35]. On the other side, sanitation is an investment, and peri-urban inhabitants made the effort to get improved toilet facilities because they had relatively consistent income to support their planning.

The other finding showed that households who live in large central regions were less likely to use improved shared toilet facilities as compared to metropolitan cities. The spatial analysis result also showed that improved shared toilet facilities were more commonly practiced in Addis Ababa and Dire Dawa, whereas the SNNPR (South Nations Nationalities and People's Region) and Beneshangul Gumuz regions were the cold spots. This is in line with a study in Mozambique that shows that shared toilet facilities are increasingly common in low-income countries in rapidly growing urban areas [1]. Moreover, in Ethiopia, as a result of internal conflict and drought, informal settlements and refugee camps that use shared toilet facilities are more common in metropolitan cities. The other possible explanation is that those large Centrals contained rural households that practiced open defecation rather than shared and improved sanitation.

The strengths of this study lie in the use of nationally representative, high-quality standardized data, which allows for generalization at the country level. However, a limitation of the study is that being cross-sectional, it cannot establish cause-and-effect relationships.

## Conclusion and recommendation

The prevalence of improved shared sanitation services usage among households in Ethiopia is relatively low. Age, educational attainment of the household head, wealth status, residence, and region were found to be significantly associated factors with improved shared sanitation facilities in Ethiopia. Moreover, there was a non-random spatial distribution of improved shared sanitation services in Ethiopia mainly found in Addis Ababa and Dire Dawa.

Based on the findings presented, it is advisable for Ethiopia to execute focused interventions aimed at rectifying the inequities in access to improved shared sanitation services, with particular emphasis on rural households and among marginalized demographics, including individuals with diminished educational and economic standing. It is essential that particular attention is directed towards regions beyond Addis Ababa and Dire Dawa, where the availability of improved sanitation services is disproportionately higher. Policymakers ought to prioritize the development of infrastructure in inadequately served regions and seamlessly integrate sanitation initiatives with comprehensive socio-economic development programs.

Ethiopia stands to gain from the adoption of analogous strategies, ensuring that interventions are characterized not only by infrastructure development but also by cultural sensitivity and community leadership. Additionally, sustained collaboration among governmental entities, non-governmental organizations, and local authorities will prove indispensable for the amplification of these initiatives. Lastly, the continuous collection of data and spatial analysis should inform decision-making processes, thereby facilitating targeted resource allocation to areas of greatest need. Further qualitative studies are needed to explore the behavioral and socio-cultural factors that may prevent individuals from utilizing improved shared sanitation facilities.

By assimilating these lessons, Ethiopia can enhance access to sanitation and improve health outcomes, drawing on the successes observed in other nations, like Nigeria and Ghana, which have higher sanitation standards.

## Author Contributions

**Conceptualization:** Baye Tsegaye Amlak, Daniel Gashaneh Belay.

**Data curation:** Daniel Gashaneh Belay.

**Formal analysis:** Baye Tsegaye Amlak, Daniel Gashaneh Belay.

**Investigation:** Baye Tsegaye Amlak, Daniel Gashaneh Belay.

**Methodology:** Daniel Gashaneh Belay.

**Software:** Daniel Gashaneh Belay.

**Supervision:** Baye Tsegaye Amlak.

**Validation:** Baye Tsegaye Amlak, Daniel Gashaneh Belay.

**Writing – original draft:** Baye Tsegaye Amlak.

**Writing – review & editing:** Baye Tsegaye Amlak, Daniel Gashaneh Belay.

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
