## [Decision Letter · Decision Letter 0]

6 Aug 2024

PONE-D-24-20290Spatial distribution and determinants of shared toilet facilities among households in Ethiopia: Using 2019 mini-Ethiopian Demographic and Health SurveyPLOS ONE

Dear Dr. Amlak,

Thank you for submitting your manuscript to PLOS ONE. After careful consideration, we feel that it has merit but does not fully meet PLOS ONE’s publication criteria as it currently stands. Therefore, we invite you to submit a revised version of the manuscript that addresses the points raised during the review process. Both reviewers have raised substantial concerns which need careful attention before the paper can be accepted.

We look forward to receiving your revised manuscript.

Kind regards,

Alison Parker

Academic Editor

PLOS ONE

4. We note that Figures 4B, 5 and 6 in your submission contain [map/satellite] images which may be copyrighted. All PLOS content is published under the Creative Commons Attribution License (CC BY 4.0), which means that the manuscript, images, and Supporting Information files will be freely available online, and any third party is permitted to access, download, copy, distribute, and use these materials in any way, even commercially, with proper attribution. For these reasons, we cannot publish previously copyrighted maps or satellite images created using proprietary data, such as Google software (Google Maps, Street View, and Earth). For more information, see our copyright guidelines: http://journals.plos.org/plosone/s/licenses-and-copyright.

1. You may seek permission from the original copyright holder of Figures 4B, 5 and 6 to publish the content specifically under the CC BY 4.0 license. 

Reviewers' comments:

Reviewer's Responses to Questions

**Comments to the Author**

1. Is the manuscript technically sound, and do the data support the conclusions?

Reviewer #1: Partly

Reviewer #2: Partly

2. Has the statistical analysis been performed appropriately and rigorously? 

Reviewer #1: I Don't Know

Reviewer #2: Yes

3. Have the authors made all data underlying the findings in their manuscript fully available?

Reviewer #1: Yes

Reviewer #2: Yes

4. Is the manuscript presented in an intelligible fashion and written in standard English?

Reviewer #1: Yes

Reviewer #2: No

5. Review Comments to the Author

Reviewer #1: NB - on quations 1 and 3 above, I don't know the answer, as I am not a statistician. There is much I cannot comment on in the statistical aspects of this work. I can only comment on the discussion of the data analysis, and whether the arguments make sense.

Thank you for the opportunity of reviewing this paper.

Overall:

1. This paper addresses a critical issue of shared sanitation and its role in reaching the goal of safely managed sanitation by 2030. Understanding the extent of shared sanitation use in Ethiopia may support strategies for realising SDG6 and for understanding what the barriers to achieving this might be.

2. I think that there is a useful paper in here, but there needs to be a bit more clarity

about what is arising from the study, what is new in the study, and what is a confirmation of other studies on the issue of shared sanitation. It also needs to be clearer when the other literature is arising from a case study in a particular country, and so may not be generalisable (will clarify below in the line by line review when this comes up). I think that there are times that other studies are used to illustrate something that does NOT arise out of this study (for example on domestic violence) and this is confusing to the reader.

3. The study could benefit from more clarity on the definitions used, particularly given the move from MDGs to SDGs and the greater focus on whether a service is safely managed rather than on the technologies used.

4. One important clarification would be whether this study assumes that people who are not using SS are practicing OD – is that correct? If so, that needs to be spelled out – that SS is the better option. Because at the moment we just read, those who use SS and those who do not, but without the clarity of what the options are of those who do not use SS. This could be my lack of statistical knowledge, but even so, it would be good to clarify in the text. To the same end, it would be useful to have a graphic of the overall breakdown of sanitation used in Ethiopia – from SMS, to SS, to OD.

5. For the results / discussion / conclusion, you could provide recommendations that arise from this study – what does the data tell you that will help planners etc. to increase access to shared sanitation, or more specifically to move people from OD to shared sanitation?

Abstract

Lns 29 -33 This starts as a good argument for including shared toilets in statistics but lines 33-34 does not clarify how understanding the prevalence of shared sanitation (SS) further supports this argument

Lns 35-50 – I am not a statistician so cannot comment

Lns 51-54 It would be useful to have a few headline results – what age or education means for the prevalence of shared sanitation

Introduction

I recommend that the discussion on the move from MDGs to SDGs happens here, and what that means for the indicators etc., as the data is from the SDG era.

e.g. lines 70-78 – are these MDG definitions, or SDG definitions?

Line 79 – check the numbers, something is not right – an increase of 3-6 million is not worth mentioning if the total using SS is 600 million

Line 80-82 – check the numbers 0.96 is not 18% of the population of SSA.

Line86 – is a word missing after ‘easy’

Lines 103-104 – what is the source of this statement (it is not SDG 6.2 as suggested by the reference)

Line 110 – they may be superior to OD – but what are the criteria?

Lines 115-117 – are you drawing a link between using , or not using SS and child mortality? Are you drawing that from your data?

Lines 118-119 – what is the relevance for planners – so that they can target particular households?

Lines 138-143 – I am confused by these numbers – there are 8663 households across 645 EAs? That doesn’t make 20-30 households per EA – please clarify here

I regret that the rest of the methods section doesn’t mean anything to me!

Results

Line 212 – can you define what a male head of household denotes and what a female head of household denotes? Is a head of household only female if there is no adult male in the house? Or how is it defined?

Line 215 I think here you could summarise up front that an increase in age, and decrease in education, wealth etc. leads to a household being more likely to use SS. Then you can go into the detail

Lines 235 – 239 Where is the risk of DV data coming from? And is SS a DV risk or a GBV risk? What is the correlation?

does this suggest a corellation? A causation? Or just two random connected indicators of increased vulnerability?

Line 244 – 247 – write this as clearly as possible also for a non-statistician audience – as I understand it, this is your main finding.

From line 306 onwards I cannot comment! Sorry!

Discussion

Again – I think it needs to be clarified what the non-SS households are using, to be clear that it is worse than SS.

Line 329 - this is not a strong reference, as it is a webpage without links to the claims it references.

Lines 331-332 Did you get this from the data in the study? Or from somewhere else (if so reference) Also add “Studies in Zambia and Ethiopia show…..” I think that the context is critical.

Line 334 – is this an assumption? |Or does education lead to households using SS (which is what your data says, I think)

Line 341 – the reference you use for the tragedy of the commons is rather old-fashioned, and the tragedy of the commons itself may not be relevant to sanitation usage. More work has been done on this more recently, e.g. by Ostrom, or by McGranahan.

Line 344 – 346 – is this speculation or from the data?

Line 346-348 it would be better to say “a study in Kenya shows….” Because otherwise it can be taken as a general truth and it may not be.

Line 347 – is ‘consistent income’ one of the indicators in the data, it’s the first time I see the qualifier ‘consistent’?

Lines 358 – 361 – this seems to be a circular argument – rewrite for clarity?

Line 361 – does ‘governments’ here refer to local governments in Ethiopia or other national governments?

Line 365 – blanket statement not support by the reference as far as I can see

Conclusion

The conclusion could usefully contain some recommendations arising from the study – how can planners / government used this data analysis to make changes to how they promote SS? What are the lessons that can be acted upon?

What still remains to be researched and analysed. What is new from this study, and what confirms previous studies?

Also – be careful what assumptions are embedded in the analysis – for example if women care so much about sanitation, why do households with a male head of household have increased access to SS?

Another still to be resolved quesiton, critical for acceptance of SS as safely managed sanitaiton is what types of shared toilet facility people are using, and whether they are properly maintained? I believe that the main issue with shared sanitation is that it is difficult to judge if they are indeed improved, or safely managed or barely better than OD.

Reviewer #2: Summary and overall impression.

The paper addresses important issue of shared sanitation facilities and puts it in the individual household and community perspective. As it uses well recognised and reliable EDHS survey data, the analysis should yield dependable results that could support policy decision makers and other stakeholders when developing interventions at local and country level. It should also provide reference points for researchers and sanitation experts alike.

From a reader point of view there is a confusion related to the definition of shared sanitation and improved sanitation. I find it difficult to see how shared sanitation is assessed in the context of the type of primary toilet type available to the household. In terms of proportions, methodology and modelling sections are broad and detailed but the outcome of the analysis that is supposed to address research questions is incomplete. Although the discussion section provides many valid points, the paper in my opinion requires major revisions to communicate its points clearly and develop more substantial recommendations that should be developed in closing (conclusions) section. Grammar, spelling mistakes and clarity of writing style must be addressed as well.

Evidence and examples

Major issues

The introduction that is to include an overview of previous research is narrow and lacks currency. It should have a clearer structure and include more recent research-based evidence. Indeed, the most recent reference cited is from 2022 (only one position). I recommend Shared sanitation in informal settlements: A systematic review and meta-analysis of prevalence, preferences, and quality. https://doi.org/10.1016/j.ijheh.2024.114392. to provide broader perspective for this paper in the context of previously published research.

Definition of shared sanitation is unclear – contradicting statements 78 vs 75.

Research question 2 (118, 119) is unclear – contributing factors (contributing to what?).

Results section needs to be re-written to address gaps (e.g. explanation of how cluster with high risk of domestic violence was identified/taken from original data set?) and improve readability by providing more detailed interpretations (see e.g. 257, 258).

Minor issues

More recent population figures are available for Ethiopia now; 2023 as opposed to cited value for 2021 (131).

Table 1 (149) does not present the logic of questions well.

Miscellaneous comments

Formatting of references requires attention – there are issues with numbers and completeness of the records.

6. PLOS authors have the option to publish the peer review history of their article (what does this mean?). If published, this will include your full peer review and any attached files.

Reviewer #1: No

Reviewer #2: No

---

## [Author Response · Author response to Decision Letter 0]

9 Sep 2024

Date: September 1, 2024

Alison Parker 

Academic Editor of Plos One Journal

Re: Spatial distribution and determinants of improved shared toilet facilities among households in Ethiopia: Using 2019 mini-Ethiopian Demographic and Health Survey (Submission ID: PONE-D-24-20290)

Dear Editor, 

We are grateful for the opportunity to revise our manuscript for further consideration for publication in Plos One Journal.

We have addressed the reviewer's comments and suggestions. Our point-by-point response describes all changes in the manuscript text. We have indicated the changes in track changes in the revised manuscript. We hope that you will find the revised manuscript acceptable for publication.

Yours sincerely,

Baye Tsegaye Amlak 

Corresponding Author 

A point-by-point response to the reviewer’s comment

We thank the editor and reviewer for their time and effort in reviewing our manuscript and for highlighting its importance. We have addressed the comments provided and revised the manuscript accordingly.

Reviewer 1

Overall: This paper addresses a critical issue of shared sanitation and its role in reaching the goal of safely managed sanitation by 2030. Understanding the extent of shared sanitation use in Ethiopia may support strategies for realising SDG6 and for understanding what the barriers to achieving this might be.

Response: Certainly, Thank you for your critical feedback and for recognizing the importance of this manuscript. We have addressed the comments provided and revised the manuscript accordingly. 

Comment #1. I think that there is a useful paper in here, but there needs to be a bit more clarity about what is arising from the study, what is new in the study, and what is a confirmation of other studies on the issue of shared sanitation. It also needs to be clearer when the other literature is arising from a case study in a particular country, and so may not be generalisable (will clarify below in the line by line review when this comes up). I think that there are times that other studies are used to illustrate something that does NOT arise out of this study (for example on domestic violence) and this is confusing to the reader.

Response: Thank you for noting this. This study focused only on improved shared sanitation and its spatial distribution in Ethiopia. Providing shared sanitation services where improved sanitation services are limited could be a crucial first step in achieving the universal sanitation coverage aim outlined in the Sustainable Development Goals. We use only variables which have been associated with the previous studies on sanitation services.

Comment #2. The study could benefit from more clarity on the definitions used, particularly given the move from MDGs to SDGs and the greater focus on whether a service is safely managed rather than on the technologies used.

Response: Thank you for noting this. The definition of the outcome variable is clearly stated in the "Study Variables" section. We have revised it based on the DHS recode manual to read: "If a household uses any of the listed improved sanitation facilities and shares them with other households it is considered as improved shared sanitation services" (see page 7, lines 153-155).

Comment #3. One important clarification would be whether this study assumes that people who are not using SS are practising OD – is that correct? If so, that needs to be spelled out – that SS is the better option. Because at the moment we just read, those who use SS and those who do not, but without the clarity of what the options are of those who do not use SS. This could be my lack of statistical knowledge, but even so, it would be good to clarify in the text. To the same end, it would be useful to have a graphic of the overall breakdown of sanitation used in Ethiopia – from SMS, to SS, to OD.

Response: Thank you for noting this. We have corrected it; accordingly, the following sentences clearly shows the outcome and controls. The outcome of this study is improved shared sanitation facilities whereas the control group are all unimproved sanitation facilities which includes OD, and all unimproved sanitation services such as Pit latrines without slab/open pits, Bucket toilets, hanging toilet/latrines and others. We modified and revised the title and all sections of the manuscript accordingly. “Spatial distribution and determinants of improved shared sanitation facilities among households in Ethiopia: Using 2019 mini-Ethiopian Demographic and Health Survey” 

The study population consisted of all houses that had unimproved sanitation services evaluated during the 2019 mini-EDHS survey (7,561). This includes, shared improved sanitation services (1,276), unimproved sanitation facilities (3,442), and open defecation (OD) (2,843). From the total 8,663, households included in the 2019 mini-EDHS survey, 1,102 households did have improved but not shared sanitation facilities and were excluded from the analysis. Ultimately, a sample of 7,561 (weighted 7,770) households was included in the analysis. (see Page 6 line no 146-151). 

Comment #4. For the results/discussion/conclusion, you could provide recommendations that arise from this study – what does the data tell you that will help planners etc. to increase access to shared sanitation, or more specifically to move people from OD to shared sanitation?

Response: Thank you for noting this. We have added a recommendations section and incorporated the issues that need to be addressed by the concerned bodies and stakeholders. Revised according to the following “The government of Ethiopia and other stakeholders should provide economic support to low-income households to construct shared sanitation facilities. Additionally, policy should emphasize user awareness and practices through education to enhance access to and usage of improved shared sanitation facilities, particularly in rural residences and large central regions. Further qualitative studies are needed to explore the behavioural and socio-cultural factors that may prevent individuals from utilizing improved shared sanitation facilities” (see page 20, lines 385-391).

Abstract

Comment #5. Line 29 -33 This starts as a good argument for including shared toilets in statistics but lines 33-34 do not clarify how understanding the prevalence of shared sanitation (SS) further supports this argument.

Response: Thank you for highlighting this important point. We have addressed the identified gap and incorporated it to support the argument. Corrected as follows: “Evidences are limited on the distribution of shared sanitation services and its determinants in Ethiopia. Therefore, this study aimed to assess the magnitude of shared toilet facilities and their determinants among households in Ethiopia.” (see Abstract page 2, lines 30-31 And page 5 line number line number 118-119 in the introduction section).

Comment #6. Lines 35-50 – I am not a statistician so cannot comment

Response: Thank you for your honest feedback, we double-checked it.

Comment #7. Lines 51-54, It would be useful to have a few headline results – what age or education means for the prevalence of shared sanitation

Response: Thank you for your valuable comment. We have corrected it as follows: “Being 55 years or older, having secondary or higher education, having middle or rich wealth status, living in urban areas, and residing in metropolitan or peripheral regions were significantly associated with the usage of shared toilet facilities.” (see page 3, lines 51-53).

Introduction

Comment #8. 

I recommend that the discussion on the move from MDGs to SDGs happens here, and what that means for the indicators etc., as the data is from the SDG era.

e.g. lines 70-78 – are these MDG definitions, or SDG definitions

Response: Thank you for your response The SDG definition states that “The Sustainable Development Goals target 6.2 calls for universal access to sanitation by 2030 and no child should die or get sick as a result of drinking contaminated drinking water, and/or being exposed to other people's excreta”

Comment #9. Line 79 – check the numbers, something is not right – an increase of 3-6 million is not worth mentioning if the total using SS is 600 million. Line 80-82 – check the numbers 0.96 is not 18% of the population of SSA

Response: Revised accordingly as “Whereas in Sub-Saharan African countries households who used shared toilet facilities climbed from 0.64 million to 0.96 million, which was 0.08% of all households” (see page 2, lines 93-94).

Comment #10. Line86 – is a word missing after ‘easy’ 

Response: We thank you. It was a grammatical and typing error. Modification has been made throughout the document.

Comment #11. Lines 103-104 – what is the source of this statement (it is not SDG 6.2 as suggested by the reference)

Response: We thank you and revised the source. https://iris.who.int/bitstream/handle/10665/258617/9789241512893-eng.pdf?sequence=1

Comment #12: Line 110 – they may be superior to OD – but what are the criteria?

Response: Thank you for raising this point: improved shared sanitation facilities are superior to open defecation for several reasons:

 They reduce the incidence of open defecation, lowering exposure to harmful pathogens and disease spread.

 They offer more privacy and dignity than open defecation, which occurs in public spaces.

 They generally cause less environmental pollution compared to open defecation, which contaminates soil and water sources.

 They provide a more structured and accessible option, especially in urban or densely populated areas.

Comment #12: Lines 115-117 – are you drawing a link between using, or not using SS and child mortality? Are you drawing that from your data?

Response: Thank you for your interesting question, it is not from our data. It is based on the reference since it is an introduction, we want to show its burden. However, we revised the child mortality section.

Comment #13: Lines 118-119 – what is the relevance for planners – so that they can target particular households?

Response: Thank you for your interesting question. Knowing the magnitude of improved shared toilet facilities and their contributing factors in Ethiopia helps program planners allocate resources more effectively, design targeted interventions, and develop relevant policies. This understanding enables better planning and decision-making, leading to improved sanitation outcomes, enhanced public health, and more efficient use of resources by addressing the specific needs and challenges faced by different communities (see page 5, lines 121-123).

Comment #14: Lines 138-143 – I am confused by these numbers – there are 8663 households across 645 EAs? That doesn’t make 20-30 households per EA – please clarify here.

Response: Thank you very much for showing the mistake we made; Now we revised accordingly, as the following “In the first stage, a total of 305 EAs (93 in urban areas and 212 in rural areas) were selected with probability proportional to EA size. The second stage was the careful selection of, on average, 25–30 homes (based on the 2019 EPHC frame)” (see page 6, lines 141-143).Though the revised sample size is 7,561, in the 2019 mini-EDHS 8663 households were included. Then 8,863/305=29 households on average.

Comment #15. I regret that the rest of the methods section doesn’t mean anything to me!

Response: No worries; thank you very much for focusing on your experts.

Results

Comment #16. Line 212 – can you define what a male head of household denotes and what a female head of household denotes? Is a head of household only female if there is no adult male in the house? Or how is it defined?

Response: Thank you for seeking clarification. Yes, as you mentioned, the sociodemographic variables of the household data/record (HR) in DHS are collected for the head of households only. Then in the Ethiopian context, mostly if both male (father) and female (mother) are present in the household, the male is considered the household head. However, if the female is leading the household alone, she is recognized as the household head. (See Table 2)

Comment #17: Line 215 I think here you could summarise up front that an increase in age, and decrease in education, wealth etc. leads to a household being more likely to use SS. Then you can go into the detail

Response: Thank you for this important comment. Revised in the manuscript as the following: “Based on the results from Model III, there was a positive association between improved shared sanitation facilities and several factors such as individuals with higher educational attainment and from greater household wealth, those living in urban areas and metropolitan regions”. (see page 13, lines 244-246).

Comment #18: Lines 235 – 239 Where is the risk of DV data coming from? And is SS a DV risk or a GBV risk? What is the correlation? does this suggest a correlation? A causation? Or just two random connected indicators of increased vulnerability?

Response: Thank you for your comments, and we apologize for any confusion. We are not entirely sure about the abbreviations "DV" and "GBV" you used. However, if "DV" refers to the dependent variable, it implies that improved shared sanitation facilities in our context. The risk associated with the dependent variable is influenced by the cluster in which individuals reside. Since we used Enumeration Area (EA) or cluster as a random variable, the variations observed in the null model (e.g., ICC, MOR) were not due to random chance but rather to the variability of the random variable, which was the cluster or enumeration area. This indicates there was a correlation or association between the random variable (cluster) and the dependent variable. This is accounted for by our multilevel analysis.

Comment #19: Line 244 – 247 – write this as clearly as possible also for a non-statistician audience – as I understand it, this is your main finding.

Response: Thank you for your suggestion: Modifications and clarification have been made in the manuscript as follows: “Based on the results from Model III, there was a positive association between improved shared sanitation facilities and several factors such as individuals with higher educational attainment and from greater household wealth, those living in urban areas and metropolitan regions”. (see page 13, lines 244-246).

Comment #20: From line 306 onwards I cannot comment! Sorry!

Response: No worries; thank you very much for focusing on your experts.

Discussion

Comment #21: Again – I think it needs to be clarified what the non-SS households are using, to be clear that it is worse than SS.

Response: Thank you for your suggestion: We have corrected it; accordingly, the following sentences show the outcome and controls. The outcome of this study is improved shared sanitation facilities whereas the control group are all unimproved sanitation facilities which includes OD, and all unimproved sanitation services such as Pit latrines without slab/open pits, Bucket toilets, hanging toilet/latrines and others. (see table 1).

Comment #22: Line 329 - this is not a strong reference, as it is a webpage without links to the claims it references.

Response: Thank you for your suggestion: revised it accordingly. 

Comment #23: Lines 331-332 Did you get this from the data in the study? Or from somewhere else (if so reference) Also add “Studies in Zambia and Ethiopia show…..” I think that the context is critical.

Response: We have got it from previous studies, and we cited references accordingly as the following. “Moreover, a higher level of education enhances awareness and fosters positive attitudes towards choosing relatively more upgraded sanitation facilities [36]. Specifically, educated women are more likely to prefer safe and sanitary facilities that offer privacy and maintain good quality during their menstrual cycle [37]. Moreover, well-educated households with higher incomes had better access to upgraded sanitation facilities [38]. ” (see page 18, lines 330-335).

Comment #24: Line 334 – is this an assumption? |Or does education lead to households using SS (which is what your data says, I think).

Response: Thank you for your valuable comments. Revised accordingly and removed the OD from the revised manuscript. Our data says 

---

## [Decision Letter · Decision Letter 1]

13 Dec 2024

PONE-D-24-20290R1Spatial distribution and determinants of improved shared sanitation facilities among households in Ethiopia: Using 2019 mini-Ethiopian Demographic and Health SurveyPLOS ONE

Dear Dr. Amlak,

Thank you for submitting your manuscript to PLOS ONE. After careful consideration, we feel that it has merit but does not fully meet PLOS ONE’s publication criteria as it currently stands. Therefore, we invite you to submit a revised version of the manuscript that addresses the points raised during the review process. Apologies for the delay in getting this back to you, one of the reviewers was not available for a while.   Both reviewers have some minor comments that still need to be addressed before we can accept the manuscript.

We look forward to receiving your revised manuscript.

Kind regards,

Alison Parker

Academic Editor

PLOS ONE

Journal Requirements:

Reviewers' comments:

Reviewer's Responses to Questions

**Comments to the Author**

1. If the authors have adequately addressed your comments raised in a previous round of review and you feel that this manuscript is now acceptable for publication, you may indicate that here to bypass the “Comments to the Author” section, enter your conflict of interest statement in the “Confidential to Editor” section, and submit your "Accept" recommendation.

Reviewer #1: All comments have been addressed

Reviewer #3: (No Response)

2. Is the manuscript technically sound, and do the data support the conclusions?

Reviewer #1: Yes

Reviewer #3: Yes

3. Has the statistical analysis been performed appropriately and rigorously? 

Reviewer #1: I Don't Know

Reviewer #3: Yes

4. Have the authors made all data underlying the findings in their manuscript fully available?

Reviewer #1: Yes

Reviewer #3: Yes

5. Is the manuscript presented in an intelligible fashion and written in standard English?

Reviewer #1: Yes

Reviewer #3: No

6. Review Comments to the Author

Reviewer #1: (No Response)

Reviewer #3: Most of the comments were addressed correctly. Here are additional comments with regards to revision1:

1) In the response to reviewers' comments you gave 3 research questions: " 1. What was the

magnitude of improved shared sanitation facilities in Ethiopia? 2. What do the spatial

distributions of improved shared sanitation facilities in Ethiopia look like? 3. What are

the associated factors with improved shared sanitation facilities?" In the text you have only 2 questions 119-121.

2) In your Multilevel model results you state:

"A household

236 from a cluster with a high risk of domestic violence had 28.75 times higher odds of having shared

237 toilet facilities than a household from a cluster with a lower risk of shared toilet facilities, according

238 to the MOR value in the null model (28.75), on the other hand, if you randomly selected two

239 households from two different clusters."

You can see you are referring to a cluster with high risk of domestic violence. How this cluster characteristics has been arrived to? Is it one of the characteristics of EA? This needs to be clarified before you report your results.

3) There is a need for further proof reading of the text. See e.g. the paragraph copied above for style and clarity and 241 (valve - should be value?).

4) Referencing - also requires additional check to make sure all statements made are link to this study results or specific source(s) see e.g. 322.

5) Recommendations are basic and could have been developed better taking into account findings and discussion. (E.g. if other countries mentioned have higher standards of sanitation are there any lessons to be learned from their experiences?

7. PLOS authors have the option to publish the peer review history of their article (what does this mean?). If published, this will include your full peer review and any attached files.

Reviewer #1: No

Reviewer #3: No

---

## [Author Response · Author response to Decision Letter 1]

15 Dec 2024

Date: December 14, 2024

Alison Parker 

Academic Editor of Plos One Journal

Re: Spatial distribution and determinants of improved shared toilet facilities among households in Ethiopia: Using 2019 mini-Ethiopian Demographic and Health Survey (Submission ID: PONE-D-24-20290)

Dear Editor, 

We are grateful for the opportunity to revise our manuscript for further consideration for publication in Plos One Journal.

We have addressed the reviewer's comments and suggestions. Our point-by-point response describes all changes in the manuscript text. We have indicated the changes in track changes in the revised manuscript. We hope that you will find the revised manuscript acceptable for publication.

Yours sincerely,

Baye Tsegaye Amlak 

Corresponding Author 

A point-by-point response to the reviewer’s comment

Reviewer #3: 

Most of the comments were addressed correctly. Here are additional comments with regards to revision1:

Thank you for your feedback and for recognizing the importance of this manuscript. We have addressed the comments provided and revised the manuscript accordingly. 

Comment #1. In the response to reviewers' comments, you gave 3 research questions: " 1. What was the magnitude of improved shared sanitation facilities in Ethiopia? 2. What do the spatial distributions of improved shared sanitation facilities in Ethiopia look like? 3. What are

the associated factors with improved shared sanitation facilities?" In the text you have only 2 questions 119-121.

Response: Thank you for noting this. In the revised manuscript, we have included three research questions? 1. To determine magnitude, 2. Spatial distribution, 3. determinants or factors. We have revised it as a short description as the following. “This study aims to address the following research questions: What is the magnitude and spatial distribution of improved shared sanitation facilities in Ethiopia? What factors are associated with improved shared sanitation facilities? See page 5 line number 119. Furthermore, those research questions were clearly answered and indicated in the manuscript (Magnitude of improved shared sanitation(Page 12 line number 239), factors associated with it(page 15 table 4), spatial distribution page 16 line number 302). 

Comment #2. In your Multilevel model results you state: "A household from a cluster with a high risk of domestic violence had 28.75 times higher odds of having shared toilet facilities than a household from a cluster with a lower risk of shared toilet facilities, according to the MOR value in the null model (28.75), on the other hand, if you randomly selected two households from two different clusters. "You can see you are referring to a cluster with high risk of domestic violence. How these cluster characteristics has been arrived to? Is it one of the characteristics of EA? This needs to be clarified before you report your results.

Response: We thank the reviewer. “Domestic violence” was the wrong word incorporated in the document. We have revised the document accordingly. Here is the comment. “According to the MOR value in the null model (28.75), a household from a high-risk cluster had 28.75 times higher odds of having shared toilet facilities compared to a household from a low-risk cluster”. Please, see line numbers 238-240.

Comment #3. There is a need for further proofreading of the text. See e.g. the paragraph copied above for style and clarity and 241 (valve - should be value?).

Response: Thank you for noting this. We have corrected the specified spelling. Moreover, we have proofread all sections of the documents.

Comment #4. Referencing - also requires additional check to make sure all statements made are linked to this study results or specific source(s) see e.g. 322.

Response: Thank you for your comment. Even though the EDHS report reported the magnitude of open defecation, we have revised and changed it with the updated articles.

“Spatiotemporal distribution and determinants of open defecation among households in Ethiopia: A Mixed effect and spatial analysis”

https://journals.plos.org/plosone/article?id=10.1371/journal.pone.0268342

Comment #5. Recommendations are basic and could have been developed better taking into account findings and discussion. (E.g. if other countries mentioned have higher standards of sanitation are there any lessons to be learned from their experiences?

Response: Thank you for your comment. 

We have addressed and revised based on your comments

---

## [Editor Report · Decision Letter 2]

18 Dec 2024

Spatial distribution and determinants of improved shared sanitation facilities among households in Ethiopia: Using 2019 mini-Ethiopian Demographic and Health Survey

PONE-D-24-20290R2

Dear Dr. Amlak,

We’re pleased to inform you that your manuscript has been judged scientifically suitable for publication and will be formally accepted for publication once it meets all outstanding technical requirements.

Kind regards,

Alison Parker

Academic Editor

PLOS ONE
---

## [Editor Report · Acceptance letter]

2 Jan 2025

PONE-D-24-20290R2 

PLOS ONE

Dear Dr. Amlak, 

I'm pleased to inform you that your manuscript has been deemed suitable for publication in PLOS ONE. Congratulations! Your manuscript is now being handed over to our production team.

Kind regards, 

on behalf of

Dr. Alison Parker 

Academic Editor

PLOS ONE